# Hourglass 3D CNN for Stereo Disparity Estimation for Mobile Robots

**Thai La [1] [ID], Linh Tao [1,*] [ID], Chanh Minh Tran [2] [ID], Tho Nguyen Duc [2] [ID], Eiji Kamioka [3] [ID] and Phan Xuan Tan [3] [ID]**

1   School of Mechanical Engineering, Hanoi University of Science and Technology, Hanoi 100000, Vietnam; thai.lq222184m@sis.hust.edu.vn

2   Graduate School of Engineering and Science, Shibaura Institute of Technology, Tokyo 135-8548, Japan; nb20502@shibaura-it.ac.jp (C.M.T.); nb20501@shibaura-it.ac.jp (T.N.D.)

3   Department of Information and Communications Engineering, Shibaura Institute of Technology, Tokyo 135-8548, Japan; kamioka@shibaura-it.ac.jp (E.K.); tanpx@shibaura-it.ac.jp (P.X.T.)

*   Correspondence: linh.taongoc@hust.edu.vn

**Abstract:** Stereo cameras allow mobile robots to perceive depth in their surroundings by capturing two separate images from slightly different perspectives. This is necessary for tasks such as obstacle avoidance, navigation, and spatial mapping. By utilizing a convolutional neural network (CNN), existing works in stereo cameras based on depth estimation have achieved superior results. However, the critical requirement for depth estimation for mobile robots is to have an optimal tradeoff between computational cost and accuracy. To achieve such a tradeoff, attention-aware feature aggregation (AAFS) has been proposed for real-time stereo matching on edge devices. AAFS includes multistage feature extraction, an attention module, and a 3D CNN architecture. However, its 3D CNN architecture learns contextual information ineffectively. In this paper, a deep encoder–decoder architecture is applied to an AAFS 3D CNN to improve depth estimation accuracy. Through evaluation, it is proven that the proposed 3D CNN architecture provides significantly better accuracy while keeping the inference time comparable to that of AAFS.

**Keywords:** mobile robots; depth estimation; stereo camera

## 1. Introduction

Mobile robots have witnessed a surge in popularity and find versatile applications in numerous fields [1]. One compelling use case for mobile robots is their deployment in hazardous environments, such as automated agriculture and the handling of dangerous materials, where they can replace human workers [2]. However, to ensure optimal performance, it is imperative for mobile robots to swiftly and accurately gauge the geometric attributes of their surroundings, specifically the depth information. Depth estimation plays a pivotal role in enabling mobile robots to excel in various tasks. It empowers these robots with the capability to perform obstacle detection [3], construct detailed environmental maps [4], and facilitate object recognition [5]. One of the potential solutions for depth estimation is stereo matching [6]. Stereo matching is a computer vision technique that simulates human vision by analyzing a pair of 2D images captured from slightly different viewpoints to reconstruct 3D scenes. The primary objective of stereo matching is to establish correspondences between pixels in these input 2D images and, subsequently, to compute the corresponding depth values for each pixel. This process is executed by identifying the disparity, which denotes the horizontal displacement between correspondences in the 2D images [7]. The accurate calculation of this disparity is instrumental in calculating the depth information, thereby empowering mobile robots to navigate, interact with, and operate effectively in their surroundings.

In order to accurately determine the disparity, recent studies have applied deep learning methods and achieved promising results [8]. Particularly, these works first used a

convolutional neural network (CNN) to extract features from 2D images, then concatenate them and store the disparity values between them to construct a 4D cost volume (*height* × *width* × *disparity* × *features*). Then, the 4D cost volume is input in a 3D CNN model for regularization into a 3D cost volume (*height* × *width* × *disparity*). Finally, the predicted disparity is regressed from the cost volume via a softmax operation ($\sigma$) [9].

For example, GC-Net [9] proposes to learn the context of cost volume through an encoder–decoder 3D CNN architecture. PSMNet [10] utilizes a feature extractor with a spatial pyramid pooling module and regularizes the cost volume using a 3D CNN based on stacked hourglass architecture. GA-Net [11] incorporates a 3D CNN with semiglobal matching for cost filtering. These approaches have demonstrated cutting-edge performance in stereo matching. Despite the high accuracy, when applying these methods to mobile robots, which often have low computational power, the computational cost is also a critical challenge.

As reported in [12], PSMNet [10] could only run at approximately 0.16 frames per second (fps) on an NVIDIA Jetson TX2 module. Similarly, although it has been proposed specifically for mobile robots, StereoNet [13] could only provide fewer than 2 fps on the same device. Such performances are far from the requirement for real-time applications in mobile robots, which is often a minimum of 30 fps [14].

Recently, the authors of [12] proposed attention-aware feature aggregation (AAFS) to obtain a better tradeoff between computational time and accuracy for real-time stereo matching on edge devices. The authors reported that AAFS could run at a maximum frame rate of 33 fps on low-budget devices, such as an NVIDIA Jetson TX2 module. However, the accuracy of AAFS is still limited due to the fact that it cannot efficiently exploit the contextual information of stereo images. The reason is that AAFS attempts to not increase the number of feature maps in its cascaded 3D CNN to limit the computational cost. In this case, leveraging the idea of a deep convolutional encoder–decoder, which is intended for dense prediction tasks, is a potential solution. Deep encoder–decoder tasks could reduce the computational cost by compressing the input data, then decoding the compressed data back to the input data dimension [15]. For example, a stacked hourglass based on a deep encoder–decoder consists of hourglass blocks that apply two-stride 3D convolutions to reduce the cost volume size by a factor of four [16]. This allows for an increase in feature dimensions with little impact on computational resources. Then, 3D transposed convolutions are applied to decode the volume to the original dimension.

Therefore, in this paper, we propose an improvement of AAFS by utilizing a stacked hourglass (encoder–decoder) architecture in a 3D CNN [16] to efficiently exploit the contextual information. Specifically, cost volume is processed iteratively in a top-down/bottom-up manner to better utilize global contextual information. The results demonstrate that the proposed method provides significantly better accuracy while requiring similar computational costs compared to AAFS.

The remainder of this paper is structured as follows. Section 2 provides a summary of existing works. In Section 3, the details of the disparity estimation framework with the proposed enhancement by a deep encoder–decoder 3D CNN are provided. Section 4 presents an experimental evaluation and analysis of results. Lastly, conclusions and directions for future work are presented in Section 5.

## 2. Related Work

Zbontar et al. originally introduced a CNN-based stereo-matching technique [17] whereby the similarity metric of tiny patch pairings was learned using a convolutional neural network. GCNet [9] was one of the first methods incorporating 4D cost volume, using the soft argmin operation in the disparity regression steps to obtain the best matching disparity. PSMNet [10] introduced a spatial pyramid pooling module and 3D stacked hourglass networks and yielded promising results. The authors of [18] proposed GwcNet, which is a modified 3D stacked hourglass architecture, and a combined 3D cost volume based on group-wise correlation. GA-Net [11] includes a semiglobal aggregation layer and

a local guided aggregation layer to replace several 3D convolution layers. To replace the 3D architecture, AANet [19] includes an intrascale and cross-scale cost aggregation algorithm, which can reduce inference time and maintain equivalent accuracy. On the other hand, DeepPruner [20], a coarse-to-fine approach, includes a differentiable PathMatch-based module to estimate the pruned search range of each pixel. Although 4D cost volume-based methods have achieved promising results, they operate at high computational cost and do not accommodate real-time operation on low-budget devices.

Therefore, some recent studies have focused on lightweight stereo networks based on 4D cost volumes to achieve real-time performance while maintaining competitive accuracy. These methods typically construct and aggregate 3D cost volume at low resolution to significantly reduce computational cost. For instance, StereoNet [13] is an edge-preserving refinement network that utilizes the left images as guidance to recover high-frequency details. Gu et al. [21] proposed a cascade cost volume, which consists of two stages. Cost volume at the early stage is built on a low-resolution feature map. Then, the later stage used the estimated disparity maps from the earlier stage to construct new cost volumes to apply better semantic features. This leads to a remarkable improvement in GPU memory consumption and computation time. AAFS [12] constructs a 4D cost volume by adopting a distance metric (height × width × disparity × 1). A disparity map is then computed at the lowest resolution, and disparity residuals are computed in later stages. However, its 3D CNN cannot exploit the contextual information for cost volume regularization, resulting in a limitation in estimation accuracy.

## 3. Methodology

In this study, we propose the integration of an hourglass architecture to the 3D CNN of the AAFS architecture [12] to better exploit the contextual information in stereo images. Such an enhancement helps achieve better disparity estimation accuracy while maintaining a competitive computational cost. As this paper is based on AAFS, we first introduce the AAFS architecture [12], then briefly describe the proposed hourglass-based 3D CNN. Figure 1 shows the architecture of AAFS [12]. The model consists of multistage feature extraction with an attention-aware aggregation module and a 3D CNN network for disparity and residual map prediction. Section 3.1 describes multiscale feature extraction based on [12]. Section 3.2 introduces correlation cost volume based on [12]. In Section 3.3, we introduced our proposed 3D CNN architecture. Section 3.4 describes disparity regression. Finally, Section 3.5 describes the loss function.

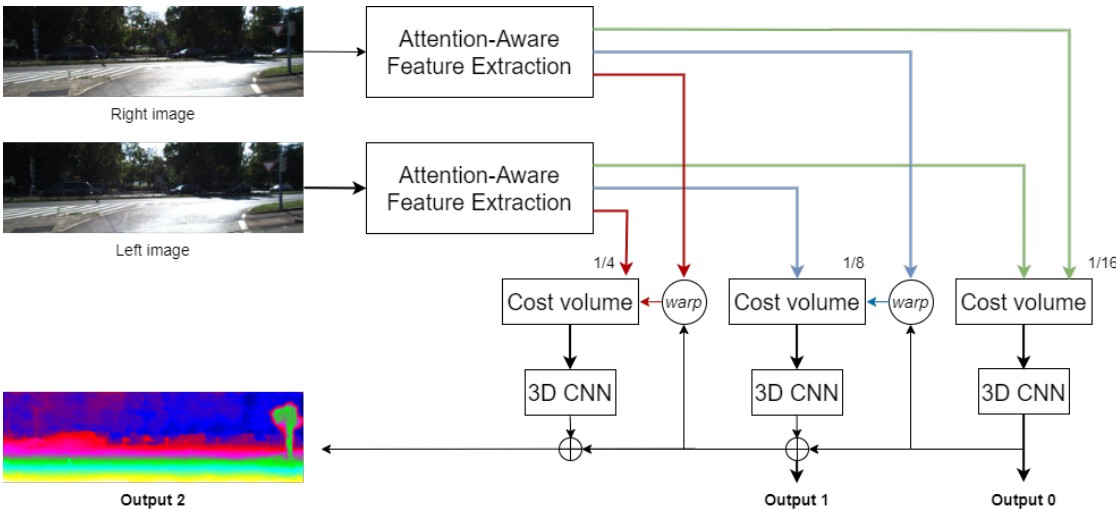

**Figure 1.** Architectural overview of AAFS.

### 3.1. Attention-Aware Feature Extraction

The authors of [12] proposed a feature extractor with an attention-aware module to obtain multistage feature maps from stereo images. Figure 2 depicts the architecture of the feature extractor, which consists of an efficient backbone, an attention module, and three feature aggregation modules.

With the attention mechanism, the feature extractor can adaptively collect input from multiple scales into encoded features. With an efficient backbone, the blueprint separable convolution (BSConv) [22] is utilized to justify DSConv [23] by analyzing intrakernel correlations of vanilla CNNs. For the purpose of multiscale representation, BSConv [22] is employed to progressively decrease the dimensions of feature maps and generate a sequence of feature maps with varying scales. Feature extraction produces output feature maps at three levels (1/4, 1/8, and 1/16). AAFS [12] includes three aggregation modules for three levels of features. Then, an attention module is employed to recalibrate channel importance and boost feature discriminability. The cost volumes for disparity estimation are formed from the final updated feature maps for each level.

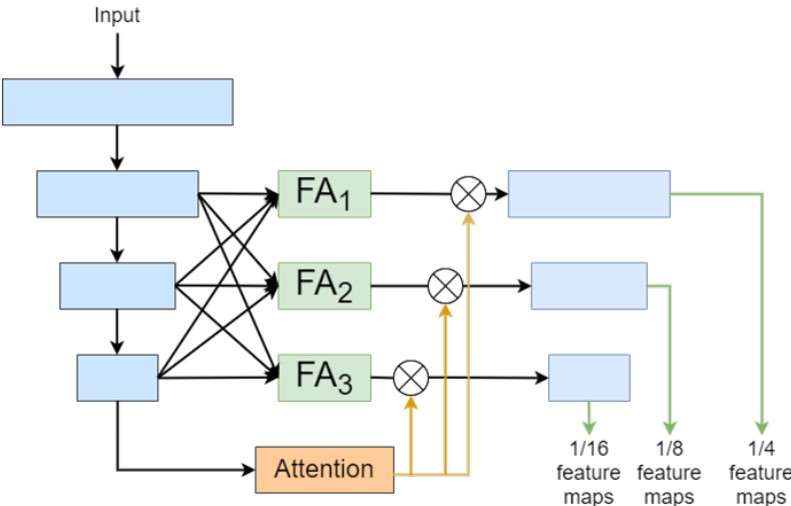

**Figure 2.** Attention-aware feature extractor.

### 3.2. Cost Volume

For real-time constraints, AAFS [12] employs a correlation volume to create a single-channel 4D cost volume for each disparity level with dimensions of $(1 \times disparity \times height \times width)$. Initially, AAFS [12] computes the full disparity map at a resolution of 1/16 in the first stage, followed by the computation of disparity residuals in later stages. This limits the search range for correspondences to 5, with offsets of $-2$, $-1$, 0, 1, and 2. To compute the residuals in later stages, AAFS [12] first scales up the rough disparity map to warp the input features. This is done by applying pixel-wise disparity estimation. Assuming that the current disparity estimation is accurate, updated right feature maps should line up with left feature maps. Estimation of residual disparity follows a similar process as the computation of the full disparity map but applying a constraint range of $-2$ to 2 for the residual disparity map. Then, the up-scaled disparity map from the earlier stage is combined with the resulting residual disparity map.

### 3.3. Hourglass 3D CNN Architecture

To aggregate the feature information from the disparity spatial dimensions, AAFS [12] includes a cascaded architecture for the 3D CNN to regularize the cost volume. Such a 3D CNN consists of five $3 \times 3 \times 3$ convolutions followed by batch normalization and ReLU, except for the final layer. Since AAFS [12] aims to achieve a tradeoff between accuracy and computational cost, it attempts to not increase the number of feature maps in the cascaded

3D CNN. In our work, an hourglass 3D CNN is proposed to replace the cascaded 3D CNN to improve the tradeoff. The architecture of the proposed 3D CNN is shown in Figure 3, with a more detailed definition of each layer proposed in Table 1. Our proposed architecture allows the model to effectively learn more contextual information, leading to accuracy improvements and maintaining a low computational cost for real-time applications on mobile devices.

As shown in Table 1, **F** features that represent the number of additional dimensions are set to 8, 4, and 4 for stages 0, 1, and 2, respectively. In a cascaded 3D CNN, the additional dimension is kept the same, with a stride of one. Thus, to increase accuracy, the additional dimension needs to be increased, which leads to a large increase in the computational cost because the same resolution is maintained. An hourglass 3D CNN is based on an encoder–decoder architecture, which reduces the required amount of computation by compressing the spatial dimension of cost volume and learning more feature dimensions, then upsampling back to the original dimension of cost volume in the decoder. Specifically, we employ two sequential $3 \times 3 \times 3$ convolutions with a stride of one followed by batch normalization and ReLU; the number of feature maps is **F** for the cost volume. Then, we compress the spatial dimension of volume by four times with two levels of the encoder. In each encoder level, two sequential $3 \times 3 \times 3$ convolutions are applied, and the stride of the first layer is two. This results in the spatial dimension of the volume decreasing by a factor of two, whereas **F** is doubled. As a result, the size of the cost volume decreases by four times, and the number of feature maps is **4F**. In addition, resolution loss decreases spatial accuracy and fine-grained details. To tackle this problem, we use two residual connections, as shown in Figure 3, to retain higher-frequency information. Then, we apply two 3D transposed convolutions with a stride of two; and the number of feature maps is 2F and F, respectively. Finally, $3 \times 3 \times 3$ convolutions in series with a stride of one and a single-feature output in the last layer are applied to predict densities. This results in a final, regularized cost volume with a size of *disparity* × *height* × *weight*.

**Table 1.** Summary of our proposed architecture. Each 3D convolutional layer represents a block of convolution, batch normalization, and ReLU (except the final layer).

| | Layer Description | Output Tensor Dim |
|---|---|---|
| | Cost volume | $1/16D \times 1/16H \times 1/16W \times 1$ |
| 1 | 3D conv, $3 \times 3 \times 3$, F features | $1/16D \times 1/16H \times 1/16W \times F$ |
| 2 | 3D conv, $3 \times 3 \times 3$, F features | $1/16D \times 1/16H \times 1/16W \times F$ |
| 3 | 3D conv, $3 \times 3 \times 3$, 2F features, stride 2 | $1/32D \times 1/32H \times 1/32W \times 2F$ |
| 4 | 3D conv, $3 \times 3 \times 3$, 2F features | $1/32D \times 1/32H \times 1/32W \times 2F$ |
| 5 | 3D conv, $3 \times 3 \times 3$, 4F features, stride 2 | $1/64D \times 1/64H \times 1/64W \times 4F$ |
| 6 | 3D conv, $3 \times 3 \times 3$, 4F features | $1/64D \times 1/64H \times 1/64W \times 4F$ |
| 7 | 3D deconv, $3 \times 3 \times 3$, 2F features, stride 2, add layer 7 and 4 | $1/32D \times 1/32H \times 1/32W \times 2F$ |
| 8 | 3D deconv, $3 \times 3 \times 3$, F features, stride 2, add layer 8 and 2 | $1/16D \times 1/16H \times 1/16W \times F$ |
| 9 | 3D conv, $3 \times 3 \times 3$, F features | $1/16D \times 1/16H \times 1/16W \times F$ |
| 10 | 3D conv, $3 \times 3 \times 3$, 1 features | $1/16D \times 1/16H \times 1/16W \times 1$ |

### 3.4. Disparity Regression

Disparity regression [9] is utilized to estimate the disparity map and is calculated as Equation (1).

$$\hat{d} = \sum_{d=0}^{D_{max}} d \times \sigma(-c_d) \tag{1}$$

where $\hat{d}$ is the predicted disparity and is calculated as the sum of each disparity probability $d$. Finally, $d$ is calculated from the cost function ($c_d$) based on a softmax operation ($\sigma(\cdot)$).

As in [9], using the above disparity regression achieves better results in comparison with stereo-matching methods based on classification.

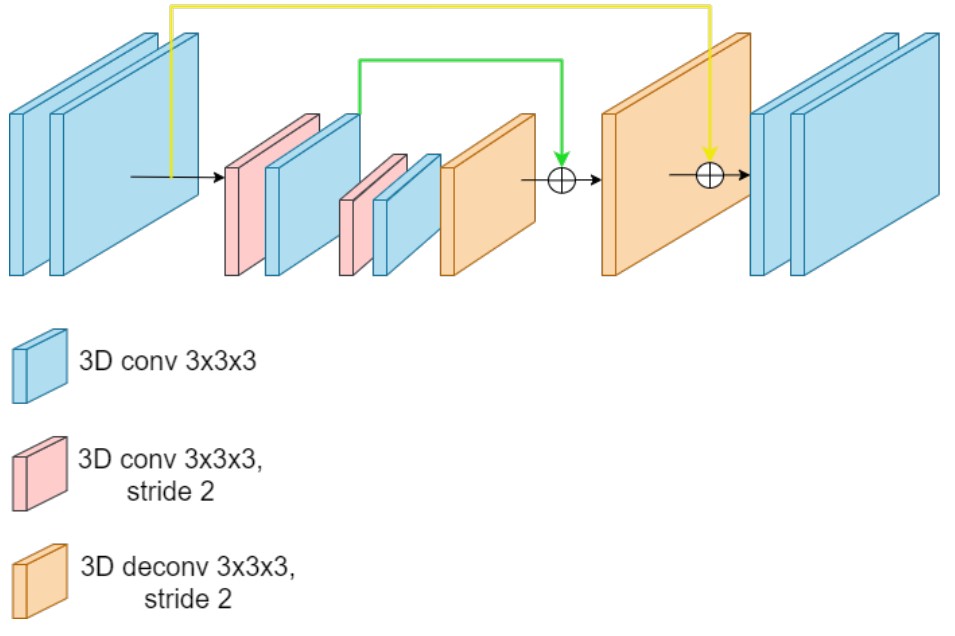

3D conv 3x3x3

3D conv 3x3x3, stride 2

3D deconv 3x3x3, stride 2

**Figure 3.** Proposed 3D CNN architecture.

*3.5. Loss Function*

In our proposal, the loss function is expressed as:

$$L = \sum_{i=0}^{i=2} \lambda_i Smooth_{L_1}\left(\hat{d}_i - d_{gt}\right) \tag{2}$$

where $d_{gt}$ is the ground truth disparity, $\lambda_i$ is the loss weight, and $\hat{d}_i$ is the predicted disparity at stage $i = 0, 1, 2$.

## 4. Experiments

*4.1. Datasets*

Figure 4 shows example scenes of the following datasets, consisting of left image, right image and ground-truth disparity.

1.  Scene Flow [24]: A large dataset with 960 × 540 pixel resolution and 35,454 training and 4370 testing stereo images. Disparity maps are presented as the ground truth. If the disparity exceeds the restrictions established in our experiment (the maximum disparity is set at 192), some pixels with large disparities are removed from the loss computation. We use the Cleanpass of the Scene Flow dataset, which contains lighting and shading effects and no additional effects.
2.  KITTI 2012 [25]: A real-world dataset containing 194 and 195 moving street view color stereo images (1240 × 376 resolution) with and without ground-truth disparities, respectively, taken from an autonomous car. In this experiment, the images with ground-truth disparities were used as training data and were split into a training set (160 images) and a validation set (34 images).
3.  KITTI 2015 [26]: A newer version of KITTI 2012, which increases the number of images both with and without ground-truth disparities to 200. Similar to KITTI 2012, the images with ground-truth disparities were used as training data and were split into a training set (160 images) and a validation set (40 images).

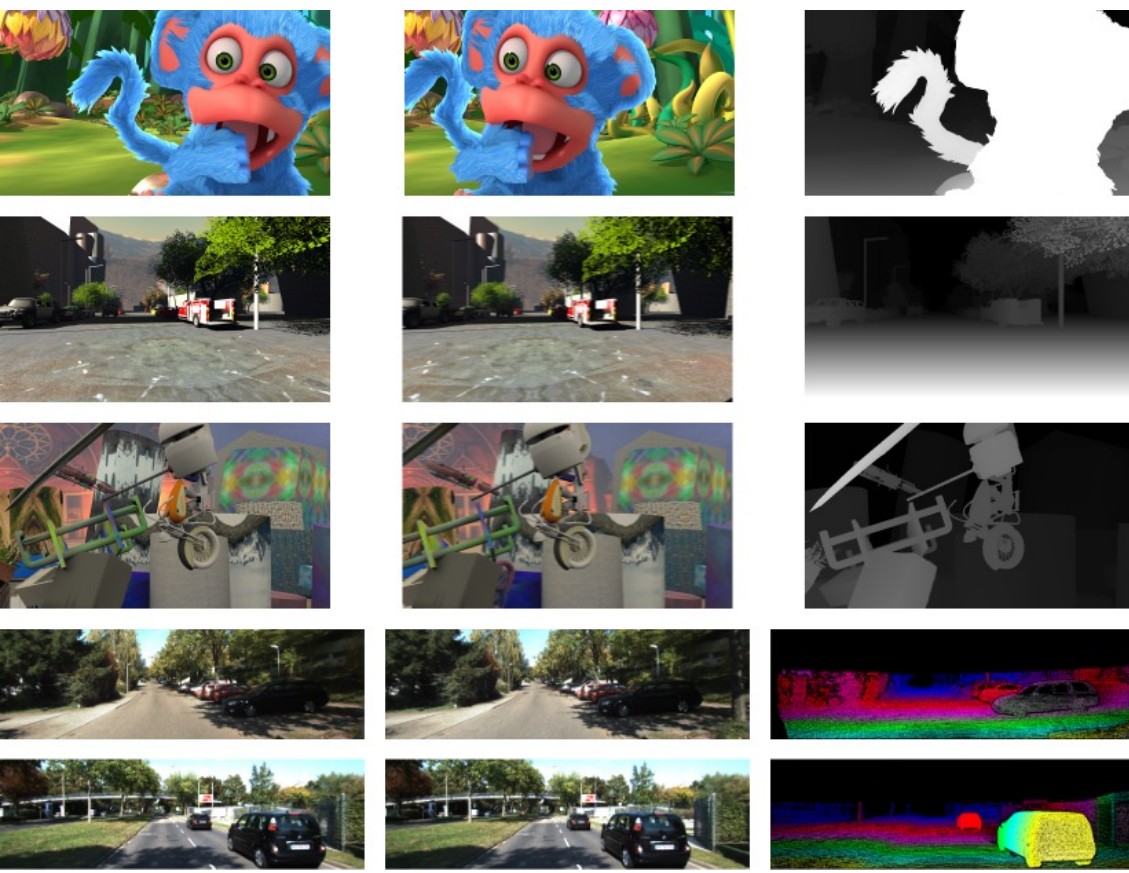

**Figure 4.** Example scenes from datasets. From top to bottom: Monkaa, Driving, FlyingThings3D in SceneFlow, and KITTI 2012 and KITTI 2015. From left to right: left image, right image, and ground-truth disparity.

*4.2. Evaluation Settings*

The proposed method was implemented on PyTorch [27]. The model was trained using an Adam optimizer with $\beta_1 = 0.9$ and $\beta_2 = 0.999$. The input images were randomly selected and randomly cropped to $512 \times 256$ size based on the random forest approach [28] during the training process. We set the maximum disparity (D) to 192.

First, we trained the model with the Scene-Flow dataset [24] for 10 epochs with a learning rate of 0.001. For evaluation on the Scene-Flow dataset, the trained model was directly used. For validation on KITTI datasets, the model trained with the Scene-Flow dataset was used for fine tuning on for 300 epochs. For the first 200 epochs, we set the learning rate of fine tuning to 0.001, and for the final 100 epochs, to 0.0001. For KITTI 2012 and 2015 submission, we combined both KITTI 2012 and KITTI 2015 training sets to fine tune the pretrained model on KITTI for 500 epochs and used the KITTI 2012 and 2015 validation sets for the KITTI 2012 and 2015 submission, respectively. For the first 300 epochs, we set the learning rate of fine tuning to 0.001, and for the final 200 epochs, to 0.0001. We set the batch size to 12 for Scene-Flow and to 6 for KITTI. Training was implemented on an NVIDIA GTX 1660. We used smooth L1 loss. We set the loss weights of the three output stages to 0.3, 0.5, and 1.0.

We evaluated the performance of our approach using two metrics. For the Scene-Flow dataset [24], we calculated endpoint error (EPE), which is the mean absolute error (MAE) between disparity prediction and ground-truth disparity. For the KITTI 2012 [25] and KITTI 2015 [26] datasets, we used three-pixel error (3px), which is the percentage of pixels with a disparity of less than three pixels or less than 5% of the ground-truth disparity. To evaluate the computational costs, we relied on floating-point operations (FLOPs). The processing speed was clocked using Google Colab (Nvidia Tesla T4) in milliseconds (ms).

The proposed method was evaluated against AAFS [12] and other state-of-the-art methods, namely PSMNet [10], DeepPruner-fast [20], and StereoNet [13]. The results are summarized and discussed in the next section.

### 4.3. Evaluation Results and Discussion

Table 2 provides the performance of the evaluated methods in terms of 3-px error (for KITTI datasets), as well as their computational cost and real-time ability (for Scene Flow dataset), while Table 3 compares the EPE (on the Scene-Flow dataset) with that of real-time models.

**Table 2.** Comparison of our proposed method with state-of-the-art accuracy-oriented methods on the KITTI 2012/2015 validation set.

| Method | KITTI 2012 (%) | KITTI 2015 (%) | FLOPs (G) | Runtime (ms) |
|---|---|---|---|---|
| PSMNet [10] | **1.39** | **1.77** | 737.84 | 1260 |
| DeepPruner-fast [20] | - | 2.59 | 208.52 | 210 |
| StereoNet-16x [13] | - | 6.32 | 59.82 | 180 |
| AAFS [12] | 6.01 | 6.76 | **0.61** | **12** |
| Ours | 5.27 | 4.42 | **0.71** | 13 |

**Table 3.** Comparison with the Scene-Flow test set.

| Method | Ours | AAFS | StereoNet-16x |
|---|---|---|---|
| EPE | **3.2** | 3.9 | 3.6 |

Overall, according to Table 2, we can observe a common tendency. The higher the accuracy, the more computational cost is required. In particular, DeepPrunner-fast [20] and PSMNet [10] produced impressively accurate results in terms of 3-px error. Those values were all less than 3%. As a result, those FLOPs and running times are the most severe. For example, DeepPruner-fast's FLOPs is 208.52 G, whereas its running time is 210 ms. From the table, we can also find the lowest accuracy in AAFS and StereoNet. However, their FLOPs and running time are relatively small. For example, StereoNet's FLOP is 59.82 G, and its running time is 180 ms. As shown in Table 3, the proposed method significantly reduced EPE by 0.7 on AAFS [12] and 0.4 on StereoNet-16x [13] on the Scene-Flow test set.

In general, our proposed method is remarkably faster than the other baselines because the disparity was computed at low resolution. Specifically, our proposal was 7.5 times faster than StereoNet-16x [13] and 14 times faster than DeepPruner-fast [20]. The proposed method outperforms StereoNet-16x [13] by a considerable margin, with only 1 billion FLOPs, clearly proving that the proposed method can accurately estimate disparity maps in real time in mobile robots.

Figure 5 compares the results acquired from the KITTI 2012 and 2015 leaderboard for PSMNet [10] and AAFS [12]. In KITTI 2012 (in Figure 5a), the error map scales linearly from 0 to >=5 pixel error; red color represents all occluded pixels that are outside the image limits. In KITTI 2015 (in Figure 5b), the error map depicts correct estimates; <3 px or <5% error are indicated in blue, and incorrect estimates are indicated in red, whereas dark regions denote the occluded pixels that lie outside the image limits. Through error maps, PSMNet [10] achieved the best results, and our proposed method improved the accuracy at the boundary of the object compared to AAFS [12]. Beyond the boundary areas, our proposed method and AAFS [12] have the same accuracy as PSMNet [10], which shows that our proposed method can accurately estimate disparity maps for real-world applications while requiring a low computational cost.

**Benchmark Results** For benchmarking, KITTI 2012/2015 test sets were used for evaluation. As shown by the online leaderboard in Table 4, the overall thee-pixel error of the proposed model was 5.64% on KITTI 2012 and 6.14% on KITTI 2015. On the other hand,

PSMNet [10], DeepPruner-fast [20], GWCNet [18], and Fast-ACVNet [29] are state-of-the-art approaches that [29] can yield accurate results; however, they cannot be applied to mobile robots due to their significant latency (>140 ms). In [29], Fast-ACVNet achieved 22–23 fps with the an NVIDIA RTX 3090 GPU, which is a challenge to deploy in mobile robots for real-time applications. In terms of the Noc metric (the proportion of incorrect pixels in non-occluded areas), our proposal improved from 6.10% to 4.93% on the KITTI 2012 test set and from 7.12% to 5.78% on the KITTI 2015 test set compared to AAFS [12]. Moreover, in terms of the All metric (the proportion of incorrect pixels in total), our proposal significantly improved from 6.94% to 5.64% on the KITTI 2012 test set and from 7.54% to 6.14% on the KITTI 2015 test set. We evaluated runtime based on Google Colab (Nvidia Tesla T4), and the results showed that our method is approximately equal to AAFS [12].

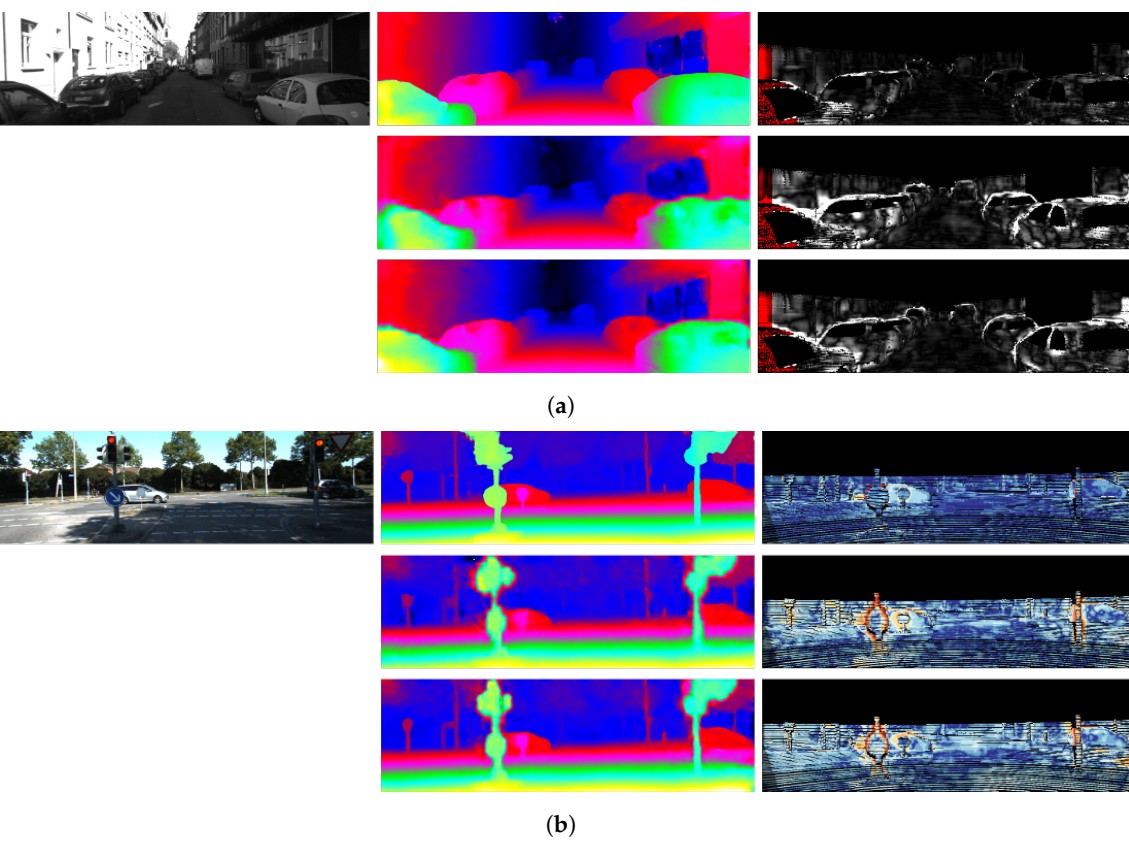

**Figure 5.** Qualitative comparisons between PSMNet, AAFS, and our proposed method. (**a**) Qualitative comparison of results on the KITTI 2012 test set. From top to bottom: PSMNet, AAFS, and our proposal. From left to right: left image, disparity prediction, and error map. (**b**) Qualitative comparisons of results on the KITTI 2015 test set. From top to bottom: PSMNet, AAFS, and our proposal. From left to right: left image, disparity prediction, and error map.

**Table 4.** Comparison on the KITTI 2012 and 2015 test benchmarks.

| Method | KITTI 2012 | | KITTI 2015 | | Params | Time |
| | Noc | All | Noc | All | (M) | (ms) |
|---|---|---|---|---|---|---|
| PSMNet [10] | 1.49 | 1.89 | 2.14 | 2.32 | 5.22 | 1260 |
| Fast-ACVNet [29] | - | - | 1.77 | 1.91 | 3.07 | 140 |
| DeepPruner-fast [20] | - | - | 2.35 | 2.59 | 7.47 | 210 |
| GwcNet [18] | 1.32 | 1.70 | 1.92 | 2.11 | 6.52 | 692 |
| DeepPruner-best [20] | - | - | 1.95 | 2.15 | 7.39 | 685 |
| AAFS [12] | 6.10 | 6.94 | 7.12 | 7.54 | 0.023 | 12 |
| Ours | 4.93 | 5.64 | 5.78 | 6.14 | 0.12 | 13 |

## 5. Conclusions

Recent studies have proven that utilizing CNNs significantly improves the performance of stereo matching. However, for application like mobile robots, it is still a challenge to estimate disparity in real time. In this work, inspired by AAFS [12], we propose the application of a 3D CNN based on a stacked-hourglass architecture to efficiently aggregate the feature information with the disparity dimension. Through experiments, the proposed method has been proven to achieve better accuracy than AAFS while requiring a similar computational cost. The predicted disparity map undoubtedly proves the ability of the proposed method to accurately estimate depth for real applications.

**Author Contributions:** Conceptualization, T.L., C.M.T. and T.N.D.; Methodology, T.L., C.M.T. and T.N.D.; Supervision, L.T., E.K. and P.X.T.; Writing—original draft, T.L., C.M.T. and T.N.D.; Writing—review and editing, L.T., E.K. and P.X.T. All authors have read and agreed to the published version of the manuscript.

**Funding:** This research received no external funding.

**Institutional Review Board Statement:** Not applicable.

**Informed Consent Statement:** Not applicable.

**Data Availability Statement:** Not applicable.

**Conflicts of Interest:** The authors declare no conflict of interest.

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
