# Peer review of "Hourglass 3D CNN for Stereo Disparity Estimation for Mobile Robots"

_applsci, doi:10.3390/app131910677_

Round 1

Reviewer 1 Report

Title: Hourglass 3D CNN for Stereo Disparity Estimation for Mobile Robots

This paper presents a stereo disparity estimation method which combines the Attention-Aware Feature Aggregation [12] and Hourglass 3D CNN techniques. The authors claim that by replacing the cascaded 3D CNN used in Attention-Aware Feature Aggregation [12] by the Hourglass 3D CNN, the proposed method can maintain real-time depth map estimation speed compared with the existing method [12], while improving its performance. They highlight the advantage of its applicability in fields that require real-time processing, such as mobile robots and phones. Depth image techniques based on stereo imaging have been enhanced with recent artificial intelligence methods like 3D CNN. Accurately and quickly estimating depth images in 3D image processing can be considered a very valuable topic to research. Thus, it is worthy of publication in this journal, but there are some concearns that need to be resolved.

1.    In Table 2, the FLOPs of AAFS and the proposed method are very close at 0.61B and 0.71B, respectively, with runtimes shown as 12ms and 13ms. However, in Table 4, the #params of AAFS and the proposed method differ greatly at 0.023M and 0.12M, yet their run times only show a minor difference at 13ms and 14ms. The number of parameters is a critical factor for computation speed and memory requirements. The slight decrease in time might be due to the implementation method of AAFS. This means that by changing the development platform or the implementation language, AAFS might show considerably faster results than the proposed method. In this case, the performance improvement effect of the proposed method will be significantly reduced. Therefore, a clear explanation on this is necessary.

2.    The schematic of the proposed method in Figure 3 is inadequate. It needs improvement to a more detailed diagram for some intuitive understanding.

3.    More detailed explanations are needed for sections 3.4 and 3.5.

4.    Why is AAFS, not AAFA, short for Attention-Aware Feature Aggregation

n/a

Reviewer 2 Report

As state by authors, mobile robots have witnessed a surge in popularity and find versatile applications in numerous fields, such as automated agriculture and the handling of dangerous materials, where they may be useful and replace human workers. In this sense, it is an interesting, current and valid work aligned with the scope of the Applied Sciences Journal. The manuscript respects the current standards for the writing of scientific articles and has been well-structured and well-organized. No English language problems haves been identified, achieved results are well-described and respond to stated objectives of the work. Therefore, in order to improve the final version of the manuscript, I ask authors to consider two, but not mandatory issues:  

1. Being stereo disparity estimation for mobile robots an emergent and a very explored area of researching, consider and update of the bibliography references.

2. Although it is outlined in the current version of the manuscript, consider to highlight better the (own) authors contributions beyond SOA.

Reviewer 3 Report

This paper presents a variation of the Attention-Aware Feature Aggregation (AAFS) proposed for real-time stereo matching. The authors introduced a deep encoder-decoder architecture to improve depth estimation accuracy, that provides better accuracy (around a 25%) while keeping the inference time comparable to AAFS.

The paper is well written, and their evaluation method are correct and well described.

Only an important detail must be clarified in table 4: Why is the parameter number much bigger in their method vs. the AAFS? (0.12 M vs. 0.023 M). Please discuss according to the additional layers.

Some details to improve the manuscript:

l. 38. This sentences sounds bad; please rewrite it. “ Finally, the disparity is regressed softmax operation”

l.141. D is not defined

Table 1: Should rows 7 , 8  be “3D deconv” instead of “3D conv” (according to fig. 3)?

Table 2: please use Greek multipliers: “FLOPs (G) or FLOPs (T)” instead of “FLOPs (B)”. same for lines 241, 243.

References: please revise the upper case letters like ieee, Ga-net, etc. ; and delete duplications like “In Proceedings of the Proceedings” e.g. in ref. 9,10,11,12,13
